# Spatial distribution, pollution, and health risk assessment of heavy metal in agricultural surface soil for the Guangzhou-Foshan urban zone, South China

**Yi Xiao**[1], **Mingyan Guo**[1], **Xiaohong Li**[1], **Xixiang Luo**[1], **Ruikang Pan**[1], **Tingping Ouyang**[1,2] *

**1** School of Geography, South China Normal University, Guangzhou, China, **2** Key Laboratory of Ocean and Marginal Sea Geology, Guangzhou Institute of Geochemistry, Chinese Academy of Sciences, Guangzhou, China

* oyangtp@m.scnu.edu.cn

## Abstract

During the past decades, heavy metal pollution in agricultural soil and its impact on human health have been becoming one of the most important global environmental problems. In this research, heavy metal (Cu, Pb, Zn, Cr, Cd, Ni, As, Hg) concentrations were measured for four hundred and two soil samples collected from agricultural area within the Guangzhou-Foshan urban zone. Soil heavy metal pollution was evaluated used geoaccumulation index and potential ecological risk index. The dose response model proposed by the USEPA was used to estimate the potential health risk caused by heavy metals in agricultural soil. The results showed that: 1) Cd and Hg were the main heavy metal pollutants in agricultural soil of the study area. 89.1% and 93.3% of total soil samples suffered medium to heavy potential ecological risk caused by Cd and Hg, respectively. 2) The THI and TCR were respectively greater than 1.0 and $1.0 \times 10^{-4}$, indicating that heavy metals in agricultural soil were likely to constitute non-carcinogenic and carcinogenic risks, both of which were mainly brought by product consumption, to the public in the study area. The non-carcinogenic risks were mainly caused by Cr and As, while the carcinogenic risks were mainly from Cr, Cd, and As.

## Introduction

As the base of food production and the sink of pollutants, soil heavy metal pollution has been proving as one of the most important environmental problems [1–5]. Ecological and public health risks caused by soil heavy metal pollution have been becoming a focus of soil environment research [6, 7]. The impact of soil heavy metal pollution on human health was well-documented [5, 8, 9].

Comparing to urban and forest soils, agricultural soil is more important for soil environment research not only because of much more exogenous factors but also due to its food production function [10–13]. Agricultural soil in cities plays an important role in providing food for urban residents and protecting ecosystems. However, heavy metal pollution of agricultural

**Funding:** This work was partially funded by the Natural Science Foundation of China (Grant No. 41977261) and the Guangzhou Science Technology and Innovation Commission (Grant No. 201707010402). The funder had no role in study design, data collection and analysis, decision to publish, or preparation of the manuscript.

**Competing interests:** The authors have declared that no competing interests exist.

soil is more serious and complex due to exposure to multiple emission sources [14]. The high heavy metal content in soil can lead to soil dysfunction, deterioration of environmental quality and decline in crop productivity. Enrichment of heavy metals through food chain will ultimately affect human health [15–17]. Besides occupational exposure, dietary intake of contaminated food has become the main way of human intake of heavy metals [18–20]. For example, soil ingestion was proved to be the main source of Pb exposure for children with elevated blood Pb levels in some areas [21, 22]. Therefore, effective evaluation of heavy metal pollution and its potential human health risk become very important and necessary for agricultural soil. The geological accumulation index proposed by Muller [23], which comprehensively considers the impact of natural and anthropic factors, has been widely used at soil heavy metal pollution assessment [24]. Comprehensively considering content, toxicity and ecological effect of heavy metals, Hakanson proposed the potential ecological hazard index method to evaluate heavy metal pollution in soils and sediments [25]. Subsequently, it was used worldwide and proved as an effective and reasonable method for soil heavy metal pollution assessment [26–28]. Moreover, the toxicity coefficients of heavy metals were improved due to environmental condition such as soil pH for different regions [29].

Particularly concern in soil environmental studies are the health risks associated with heavy metal content in soil [30]. Heavy metals such as Cd, Cr, As, Hg, Pb, Cu, Zn and Ni were listed as priority control pollutants by the United States Environmental Protection Agency [31]. In the 1980s, the US EPA developed a dose response health risk assessment model, which was modified several times and proved to be still effective [32–36] for assessing the human health risk caused by soil heavy metals. The model had been successfully applied to metal pollution in agricultural soil [37–40], mine soil [41], urban park soil [42] and road dust [43].

Many studies assessed potential human health risk exposed to toxic metals in soils and crops for different countries [44–46]. According to the first National Soil Survey conducted by the Ministry of Environmental Protection and Ministry of Land and Resources of China, concentrations of heavy metals in 82.8% of the soil samples exceeded the standard limits [47]. Furthermore, previous studies indicated that Guangdong province in South China was regarded as the key region to control soil heavy metal (HM) pollution [48]. Previous investigations of soil heavy metals including content, pollution evaluation, and spatial distribution characteristics for Guangzhou city indicated that this region was a high-background area of heavy metals with strong potential ecological risks [49–51]. Previous studies suggested that more attentions should be paid to children due to their higher sensitivity to heavy metal exposure than that of adults [52]. Zheng indicated that potential high health risk caused by Cd ingestion from Cd polluted leaf and root vegetable consumption was primarily concentrated in parts of Huizhou [53]. However, detail investigation of spatial distribution, pollution and human health risk assessment of heavy metal in agricultural soil for Guangzhou-Foshan urban zone was limited. The main purpose of the present study are (1) to investigate content and its spatial distribution of eight heavy metals (Cd, Pb, Cr, Cu, Ni, Zn, Hg, As) for farmland within the study area; (2) to evaluate geoaccumulation degree and potential ecological risk of heavy metal pollution for agricultural soil; (3) to assess potential carcinogenic and non-carcinogenic health risks caused by heavy metals for adults and children using a human health risk assessment model recommended by United States Environmental Protection Agency (USEPA).

## Materials and methods

### Study area and sample collection

The study area, Guangzhou-Foshan urban zone with a population density over 2000 persons per square kilometer, which is one of developing poles of the Guangdong Hongkong and

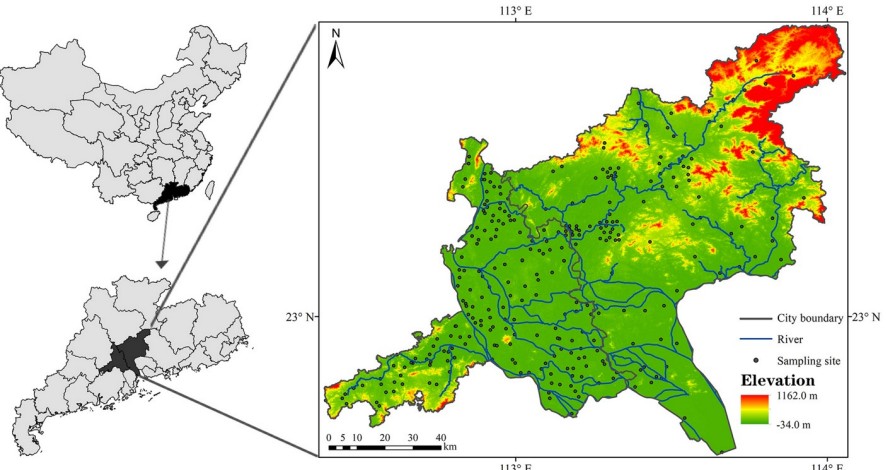

**Fig 1. Location of study area and spatial distribution of sampling sites.** The city boundary data are from the Department of Natural Resources of Guangdong Province. The GCS WGS 1984 coordinates is used to extract data through georeferencing and vectorization.

Macao greater bay area, is located at alluvial plain of the Pearl River, south China with a hot and humid subtropical monsoon climate (Fig 1). According to statistical data from the Statistics Bureau of Guangdong Province, Guangzhou-Foshan urban zone accounted for 6.25% and 33.7% for the total area and gross domestic product (GDP) of Guangdong Province in 2018, respectively [54]. Since the implementation of reforming and opening policy in China from late 1970s, Guangzhou and Foshan have been tending to be integrated at many aspects including agricultural production.

Following to the guidelines of <Specification of regional eco-geochemistry assessment> and <Determination local eco-geochemistry assessment> by the Ministry of Natural Resources of the People's Republic of China, four hundred and two surface soil (0–20 cm) samples were collected from farmland within Guangzhou-Foshan. According to distribution of farmland and terrain condition, most samples are distributed at rural and plain areas. The spatial distribution of sampling sites is shown at Fig 1.

## Heavy metal content measurements

All samples were air dried at room temperature once after collection. After removing obvious gravel and roots, the samples were ground into a powder by sieving through a 74 μm mesh for element content measurements. Following the methods in soil environmental quality standard of the People's Republic of China [55], soil samples dissolution and heavy metal content measurements were performed at the analytical and testing center of the South China University of Technology. Content of heavy metals Copper (Cu), Nickel (Ni), Zinc (Zn), Lead (Pb), Chromium (Cr), Arsenic (As), and Mercury (Hg) were determined using a Flame atomic absorption spectrophotometry, Cadmium (Cd) content were measured by a Graphite furnace atomic absorption spectrophotometry. In order to ensure the accuracy of the data and the feasibility of the results, about twenty percent of all samples were retested.

## Geoaccumulation index

Geoaccumulation index ($I_{geo}$) has been proved as an effective method for soil and sediment heavy metal contamination evaluation [56, 57]. According to Muller, the $I_{geo}$ can be calculated

for each element using the following equation [23].

$$I_{geo} = log_2 \frac{C_{sample}}{1.5 * C_{background}}$$

where $C_{sample}$ and $C_{background}$ stand for measured content in a sample and local natural background content of specific heavy metal, respectively. The number "1.5" in the equation is the coefficient of petrogenesis effect [58]. The calculated results of $I_{geo}<0$, $0 \leq I_{geo}<1$, $1 \leq I_{geo}<2$, $2 \leq I_{geo}<3$, $3 \leq I_{geo}<4$, $4 \leq I_{geo}<5$ and $I_{geo} \geq 5$ correspond to the levels of no pollution, no—medium pollution, medium pollution, medium—heavy pollution, heavy pollution, heavy—very heavy pollution and extremely heavy pollution, respectively [23, 59].

## Potential ecological risk

Comprehensively considering environmental conditions, toxicological and ecological effects of elements, the potential ecological risk index method proposed by Hakanson has been widely used for soil heavy metal pollution evaluation and pollutant identification in soil [25]. According to Hakanson, the potential ecological risk (RI) of a sample is the sum of potential ecological risk index (E) of every evaluated element. The calculation formula is [25]:

$$RI_j = \sum_{i=1}^{n} E_j^i = \sum_{i=1}^{n} T_i \times C_j^i = \sum_{i=1}^{n} T_i \times \frac{c_j^i}{c_r^i}$$

where, $RI_j$ is the comprehensive potential ecological risk index of a variety of heavy metals in sample $j$. $E_j^i$ is the potential ecological risk index of heavy metal $i$ in $j$ sample. $C_j^i$ is the measured concentration of heavy metal $i$ in a sample $j$. $c_r^i$ is the background value of heavy metal $i$; $T_i$ is the toxicity coefficient of heavy metal $i$, which represents the toxicity level of heavy metal and the sensitivity of organisms to heavy metal pollution. The heavy metal toxicity coefficients calculated by Xu et al are used in the present study [60]. The potential ecological risk level of soil heavy metals is classified according to the standards suggested by Hakanson [25]. The results $E_j^i<40$ or $RI<150$, $40 \leq E_j^i<80$ or $150 \leq RI<300$, $80 \leq E_j^i<160$ or $300 \leq RI<600$, $160 \leq E_j^i<320$ or $600 \leq RI<1200$, and $E_j^i \geq 320$ or $RI \geq 1200$ represent levels of slight risk, medium risk, strong risk, very strong risk and extremely strong risk, respectively.

## Human health risk assessment

The dose-response model for potential human health risks assessment proposed and revised by the USEPA [61, 62] was used in the present study to calculate the potential human health risks caused by the measured eight heavy metal elements. According to USEPA and previous studies, at least four exposure pathways (1) consumption of agricultural products, (2) direct and indirect ingestion of soil, e.g., through hand-to-mouth activities, (3) dermal contact, and (4) inhalation via soil vapor should be considered for agricultural soils [62, 63]. The formulas of human health risk assessment were shown in S1 File.

## Statistical analysis and spatial interpolation

A statistical software package Microsoft Excel 2013 was used for descriptive statistical analysis. Based on normality analysis of the dataset, ordinary kriging interpolation was performed using an ArcGIS version 10.2 (ESRI Inc., Redlands, CA) to acquire the spatial distribution of heavy metal content, potential ecological index and human health risk.

## Results and discussion

### Heavy metals in soil

The descriptive statistical results, as well as standard and background values and spatial distribution of the eight measured heavy metals are listed and illustrated in Table 1 and Fig 2, respectively.

It can be clearly seen from the results listed in Table 1 that the maximum content of the measured eight heavy metals are much higher than their background and standard values. Although the average heavy metal concentrations were below the national secondary standard values (Table 1), some heavy metal content in some samples exceeded the secondary standard values. In the total measured samples, 45.8%, 33.6%, 20.9%, 15.4%, 10.2%, 6.2%, 1.5%, and 0.7% of which exceeded the standard values of Cd, Hg, Cu, As, Ni, Zn, Cr, and Pb respectively. The average content of Cr, Pb, Ni, and Zn within agricultural soil in the study area were lower than their local background values (Table 1). However, the average content of Cd, Cu, Hg, and As were much higher than their background values.

From Fig 2, it's seen that relatively high content of the 8 measured heavy metals was appeared near the city boundary. From the perspective of spatial distribution, the content of Cd, Cr, Ni, Zn, and Cu ascended firstly and then descended from the northeast to the southwest of the study area (Fig 2A and 2C–2F). Although anthropogenic sources such as traffic emissions and coal combustion were considered as major sources of Pb in soil [1, 10], the high Pb content also appeared at the northeast and the southwest mountainous regions (Fig 2B), suggesting significant natural input of Pb in this region [65]. The Hg content reached the highest value in the built-up area of Foshan city, which was close to the city boundary, and then gradually decreased in the southwest region. On the other hand, the Hg content appeared relatively low in most regions of Guangzhou city (Fig 2G). These results were similar to that of the previous study by Lin et al [66]. According to previous investigation and analysis of the sources of heavy metal elements in the soils in the Pearl River Delta, the Hg content in soils was greatly affected by human activities, followed by the Hg content in typical mines. The boundary between Guangzhou and Foshan, a densely populated and economically developed area, showed an abnormal Hg content [67]. As for As content, the highest value was appeared in the southeast plain region, and then dropped to the center and the northeast region of the study area. The declining value might be caused by released pesticides [10]. The As content appeared relatively low in the southwest mountainous region (Fig 2H).

**Table 1. A statistical summary of soil heavy metal content (mg/kg).**

|  | Cd | Pb | Cr | Cu | Ni | Zn | Hg | As |
|---|---|---|---|---|---|---|---|---|
| Maximum | 1.37 | 2446.9 | 391.8 | 290 | 442 | 500.4 | 3.14 | 203.64 |
| Minimum | 0.001 | 12.28 | 3.124 | 2.368 | 2.295 | 14.3 | 0.015 | 1.91 |
| Mean | 0.2 | 34.84 | 28.51 | 19.3 | 12.02 | 45.71 | 0.26 | 16.88 |
| Standard deviation | 0.26 | 170.99 | 42.41 | 28.74 | 24.69 | 67.22 | 0.44 | 30.37 |
| Variable coefficient | 1.26 | 4.91 | 1.49 | 1.49 | 2.05 | 1.47 | 1.7 | 1.8 |
| Standard value [a] | 0.3 | 250 | 150 | 50 | 40 | 200 | 0.3 | 30 |
| Background value [b] | 0.056 | 36.0 | 50.5 | 17.0 | 14.4 | 47.3 | 0.078 | 8.90 |

a) The standard values are from the soil environmental quality standard of the People's Republic of China (GB15618-1995) [55] for dry land under second level with pH < 6.5.

b) Background values are from Handbook of Data for Environmental Background Values of Guangdong [64].

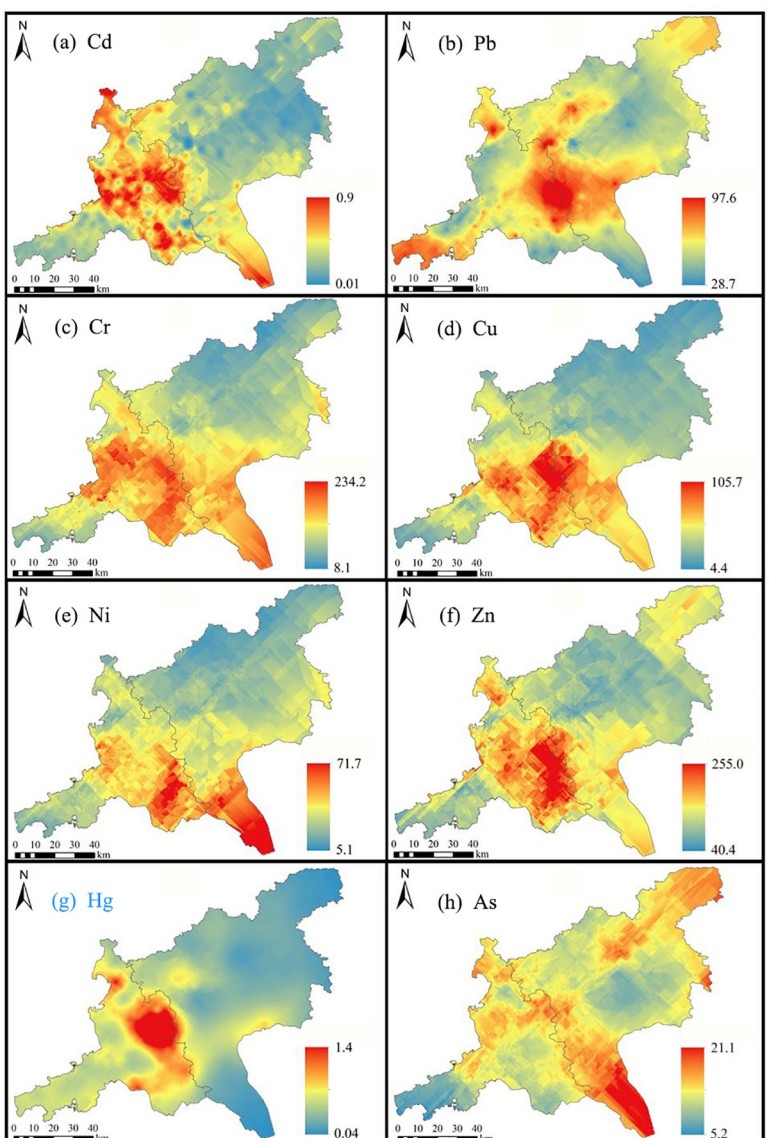

**Fig 2. Spatial distribution of heavy metal content in soil.** (a) Cd, (b) Pb, (c) Cr, (d) Cu, (e) Ni, (f) Zn, (g) Hg, (h) As.

## Evaluation of heavy metals pollution

The calculated results of the geoaccumulation index and potential ecological risk index are illustrated in Fig 3.

From Fig 3A, it is clear that cumulative levels of Pb, Cr, and Ni were relatively light. More than 90% of the measured samples were within non-pollution to no-medium pollution. However, medium, heavy, even extremely heavy pollution appeared at some samples for the other heavy metals. Less than 20% of the measured samples were medium to heavily polluted by Cu, Zn, and As (Fig 3A). More than 50% and 11% of the measured samples were medium-heavily and heavily polluted by Cd, respectively. Meanwhile, medium-heavy Hg pollution accounted for about 40% of all samples, indicating a significant anthropogenic input to the environment [66]. Furthermore, some samples were appeared heavy-very heavy Hg pollution (Fig 3A).

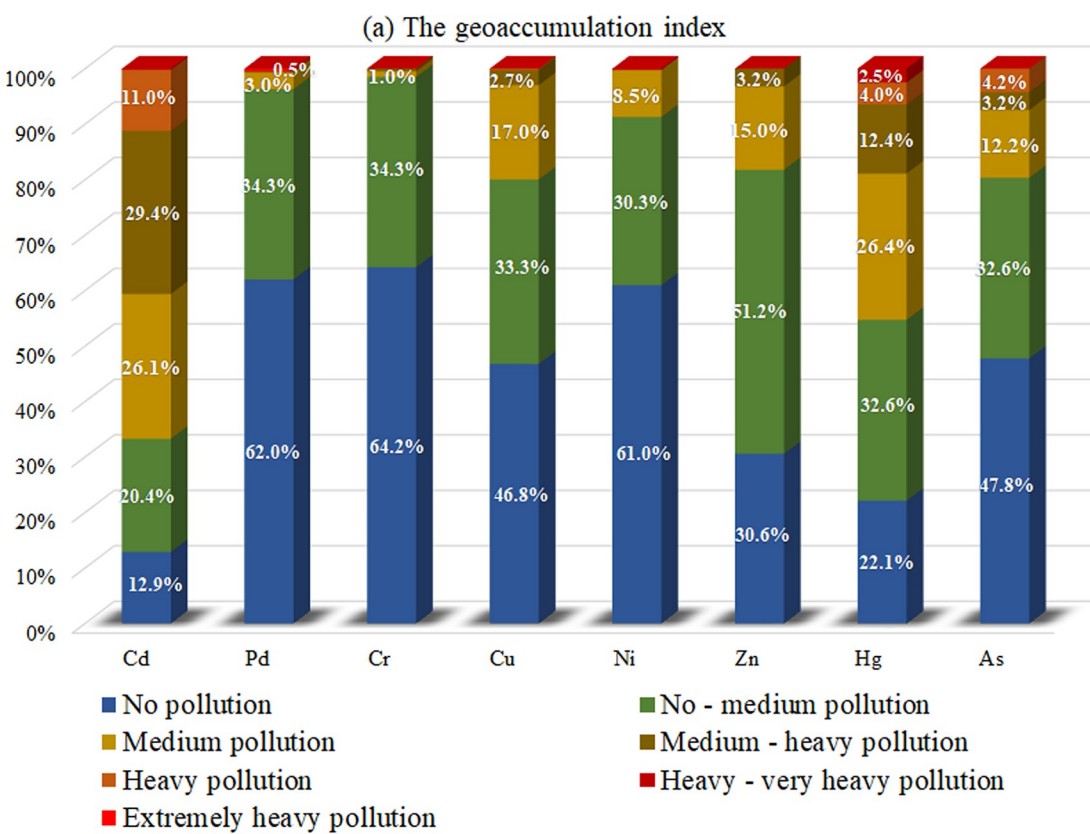

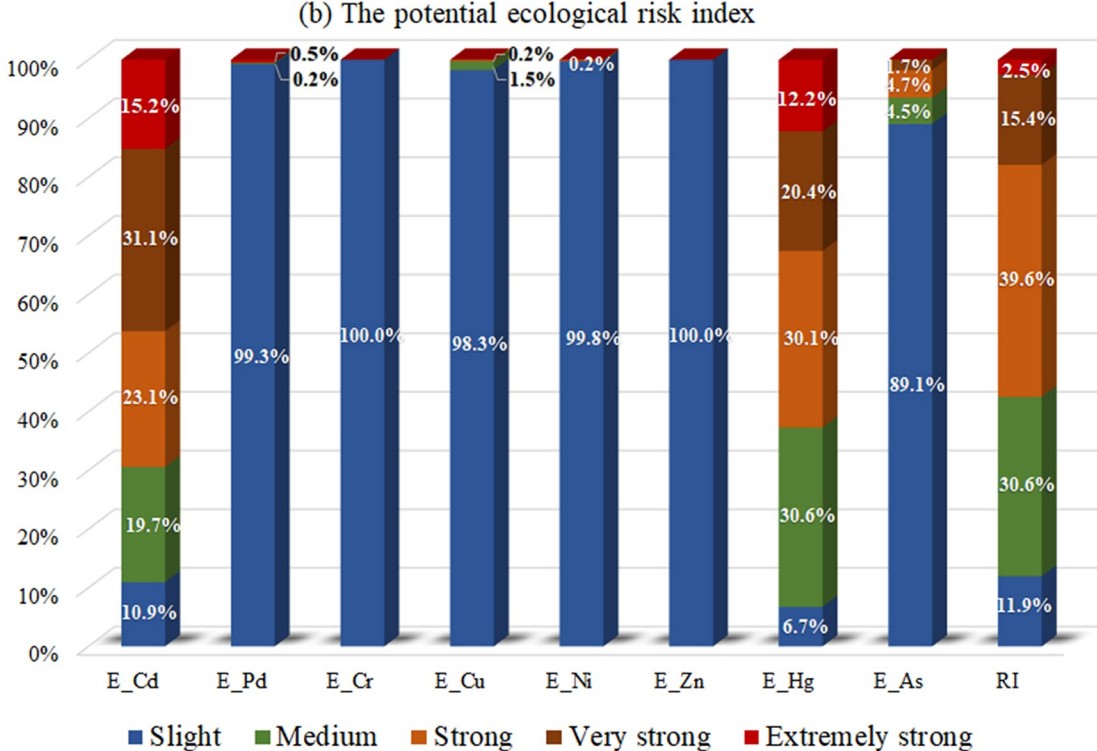

**Fig 3.** Composition of pollution level assessed due to calculated results of (a) geo-accumulation index and (b) potential ecological risk.

From the results of the potential ecological risk index illustrated in Fig 3B, the ecological risk of Pb, Cr, Cu, Ni, and Zn could be ignored since nearly all measured samples belonged to the range of slight ecological risk. Only about 10.2% of the measured samples were within the medium to the strong potential ecological risk of As. More than half of samples, however, were in the strong to the very strong potential ecological risk caused by Cd and Hg. This results was the same as the study of Bai and Liu [68]. In addition, 15.2% and 12.2% of the measured samples suffered extremely high ecological risk caused by Cd and Hg, respectively. Combined with the total potential ecological risk of the eight heavy metals, the RI results indicated that the percentages of the measured samples under medium, strong, very strong, and extremely strong ecological risks in the study area were 30.6%, 39.6%, 15.4%, and 2.5%, respectively (Fig 3B).

The impact of soil heavy metal pollution on ecology in the study area was proved to be more complex than that in other regions due to acidic soils [69, 70]. Since the main potential ecological risk of heavy metal pollution in the study area was mainly from Cd and Hg, the potential ecological risk index of Cd and Hg as well as the total potential ecological risk index (RI) were spatially interpolated for the study area using kriging interpolation with an ArcGIS 10.2, and the results are illustrated at Fig 4.

Slight and medium potential ecological risks of Cd were only distributed in a small part of the northeast of the study area. Most regions of the study area were under strong, very strong, and even extremely strong potential ecological risks of Cd (Fig 4A). It should be pointed out that very strong and extremely strong potential ecological risks of Cd were mainly distributed in southeast of Guangzhou and almost the whole Foshan city (Fig 4A), which was the same as the distribution of the Cd content. As for potential ecological risk of Hg, extremely strong potential ecological risk concentrated in Foshan built-up boundary close to Guangzhou (Fig 4B), which was similar with the distribution of the high content of Hg. The very strong

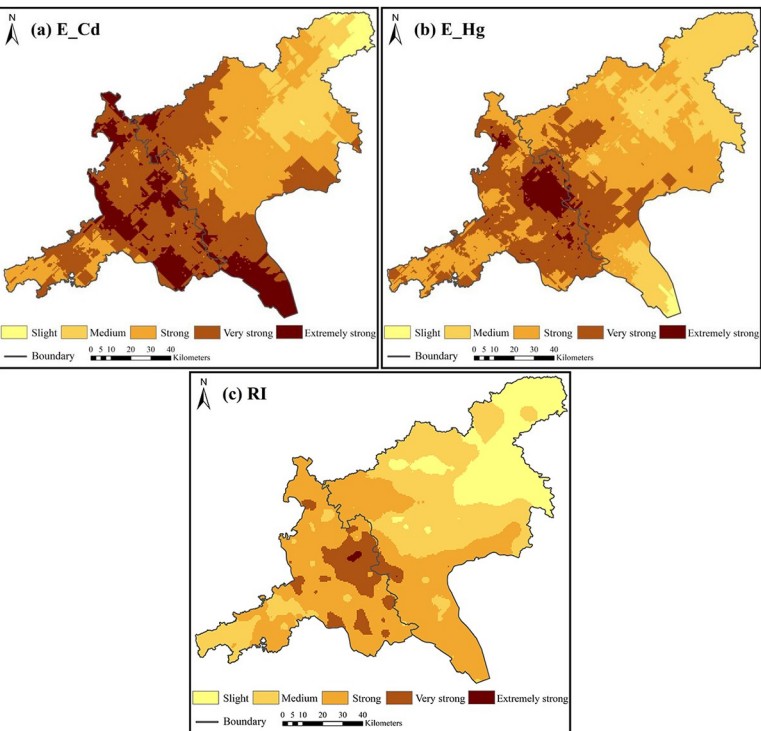

**Fig 4.** Spatial distribution of potential ecological risk of (a) $E_{Cd}$, (b) $E_{Hg}$, and (c) RI.

potential ecological risk was distributed in the junction of Guangzhou and Foshan, and mainly in the territory of Foshan. The strong potential ecological risk of Hg was distributed in the southwest Foshan as well as the central and the eastern Guangzhou. Medium and slight potential ecological risks of Hg were appeared in small regions of the northeast and the southeast of the study area (Fig 4B).

According to the spatial distribution of RI illustrated in Fig 4C, very strong and extremely strong potential ecological risks were scattered in Foshan city and concentrated in a very small region in Foshan built-up areas. Most central and northeast regions of Guangzhou city and southwest region of Foshan city were under slight to medium potential ecological risk. The strong potential ecological risk was occupied in most regions of Foshan city (Fig 4C).

## Assessment of potential human health risk

According to the USEPA, combined with some previous studies, HQ and THI values less and greater than 1.0 indicated negligible and considerable human health risk of developing chronic diseases, respectively [4, 71, 72]. As for carcinogenic risk (CR), values $1.0 \times 10^{-6}$ and $1.0 \times 10^{-4}$ were the thresholds of safety level and the maximum tolerable risk, respectively [5]. The non-carcinogenic and carcinogenic health risks, caused by heavy metals in soil, were estimated to be exposed through the consumption of agricultural products, as well as the direct and indirect ingestion, dermal contact, and inhalation via soil vapor were listed in Table 2 for adults and children. Furthermore, the contributions of each element to THI and TCR were illustrated in Fig 5.

Generally, children were under higher non-carcinogenic and lower carcinogenic health risks than adults because children presented higher THI values and lower TCR values (Table 2). The similar results could also be found in previous findings of Chen et al. [48]. The results of THI were greater than 1.0 for both adults and children. Furthermore, the TCR values exceeded the maximum tolerable risk ($1.0 \times 10^{-4}$) for both adults and children (Table 2). These results indicated that both adults and children in the study area were likely to experience non-carcinogenic and carcinogenic risks.

**Table 2. Estimation of non-carcinogenic (hazard quotient, HQ) and carcinogenic health risks (carcinogenic risk, CR) caused by soil heavy metals.**

| | Adults (aged 18-) | | | | | Children (aged 0–17) | | | | |
|---|---|---|---|---|---|---|---|---|---|---|
| | Soil ingestion | Products consumption | Dermal contact | Air inhalation | Total pathways | Soil ingestion | Products consumption | Dermal contact | Air inhalation | Total pathways |
| **HQ** | | | | | | | | | | |
| Cd | 1.1E-04 | 1.5E-01 | 2.5E-04 | 1.1E-06 | 1.5E-01 | 6.4E-04 | 2.9E-01 | 1.4E-03 | 1.3E-06 | 2.9E-01 |
| Pb | 6.3E-03 | 1.8E-01 | 2.4E-03 | 3.7E-06 | 1.9E-01 | 3.6E-02 | 3.5E-01 | 1.4E-02 | 4.0E-06 | 4.0E-01 |
| Cr | 7.1E-03 | 9.6E-01 | 2.0E-02 | 4.4E-04 | 9.9E-01 | 4.1E-02 | 1.9E+00 | 1.1E-01 | 4.8E-04 | 2.1E+00 |
| Cu | 2.9E-04 | 3.7E-01 | 5.5E-05 | 1.7E-07 | 3.7E-01 | 1.7E-03 | 7.3E-01 | 3.1E-04 | 1.9E-07 | 7.3E-01 |
| Ni | 3.6E-04 | 1.0E-01 | 5.2E-04 | 1.6E-07 | 1.0E-01 | 2.1E-03 | 2.0E-01 | 2.9E-03 | 1.8E-07 | 2.0E-01 |
| Zn | 1.2E-04 | 2.1E-01 | 3.5E-05 | 7.1E-08 | 2.1E-01 | 7.0E-04 | 4.1E-01 | 2.0E-04 | 7.8E-08 | 4.1E-01 |
| Hg | 7.4E-04 | 3.1E-01 | 4.2E-05 | 8.1E-07 | 3.1E-01 | 6.8E-03 | 6.1E-01 | 2.4E-04 | 8.9E-07 | 6.2E-01 |
| As | 2.6E-02 | 8.1E-01 | 1.5E-03 | 1.5E-05 | 8.3E-01 | 1.5E-01 | 1.6E+00 | 8.3E-03 | 1.7E-05 | 1.8E+00 |
| Total | 4.1E-02 | 3.1E+00 | 2.5E-02 | 4.6E-04 | 3.2E+00 | 2.4E-01 | 6.1E+00 | 1.4E-01 | 5.0E-04 | 6.5E+00 |
| **CR** | | | | | | | | | | |
| Cd | 2.1E-07 | 2.8E-04 | 1.20E-08 | 1.30E-10 | 2.8E-04 | 3.1E-07 | 1.4E-04 | 1.7E-08 | 3.6E-11 | 1.4E-04 |
| Cr | 3.3E-06 | 4.5E-04 | 7.60E-06 | 1.60E-07 | 4.6E-04 | 4.8E-06 | 2.2E-04 | 1.1E-05 | 4.5E-08 | 2.4E-04 |
| **As** | 3.6E-06 | 1.1E-04 | 5.00E-07 | 2.10E-08 | 1.2E-04 | 5.2E-06 | 2.2E-04 | 7.2E-07 | 5.9E-09 | 2.3E-04 |
| Total | 7.2E-06 | 8.4–04 | 8.12E-06 | 1.86E-07 | 8.6E-04 | 1.0E-05 | 5.6E-04 | 1.2E-05 | 5.1E-08 | 6.1E-04 |

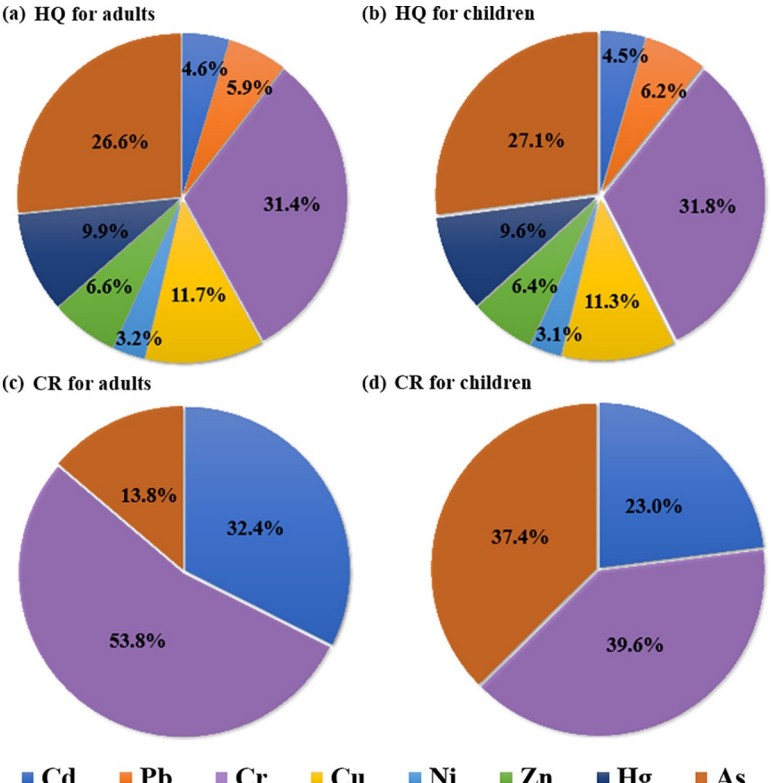

**Fig 5. Contribution of non-carcinogenic and carcinogenic caused by each element to THI and TCR.** HQ for (a) adults and (b) children, CR for (c) adults and (d) children.

From Table 2, it could be clearly seen that both non-carcinogenic and carcinogenic risk exposed through pathway product consumption accounted for more than 90% of the THI and TCR for both adults and children. Moreover, HQ and CR exposed through pathway products consumption were much greater than 1.0 and 1×10$^{-4}$, for both adults and children (Table 2), indicating that products consumption might bring significant potential non-carcinogenic and carcinogenic health risks to the public of the study area [4]. As for the contribution of each element for the THI and TCR, the HQ values of the calculated eight heavy metals for both adults and children were ranked as: Cr > As > Cu > Hg > Zn > Pb > Cd > Ni (Table 2). The numerical results were basically consistent with the research of Zheng et al. [53]. The estimated HQ values of the measured 8 heavy metals through all exposed pathways were less than 1.0 for adults. However, the HQ values of heavy metals Cr and As through products consumption were greater than 1.0 for children (Table 2). Furthermore, the HQ values of Cr and As accounted for more than 55% of the THI for both adults and children, respectively (Fig 5A and 5B). These results indicated that potential non-carcinogenic health risks were mainly caused by Cr and As though products consumption, which was consistent with previous studies [4]. The CR values of the calculated three heavy metals exposed through product consumption were greater than the maximum tolerable risk (1×10$^{-4}$) for both adults and children (Table 2), indicating that these heavy metals might bring potential carcinogenic health risk to the public in the study area through products consumption. In order to figure out the specific regions of the study area where experienced potential health risks, the spatial interpolation results of the estimated HQ values of Cr and As, as well as CR values of Cr, Cd, and As were illustrated in Figs 6 and 7, respectively.

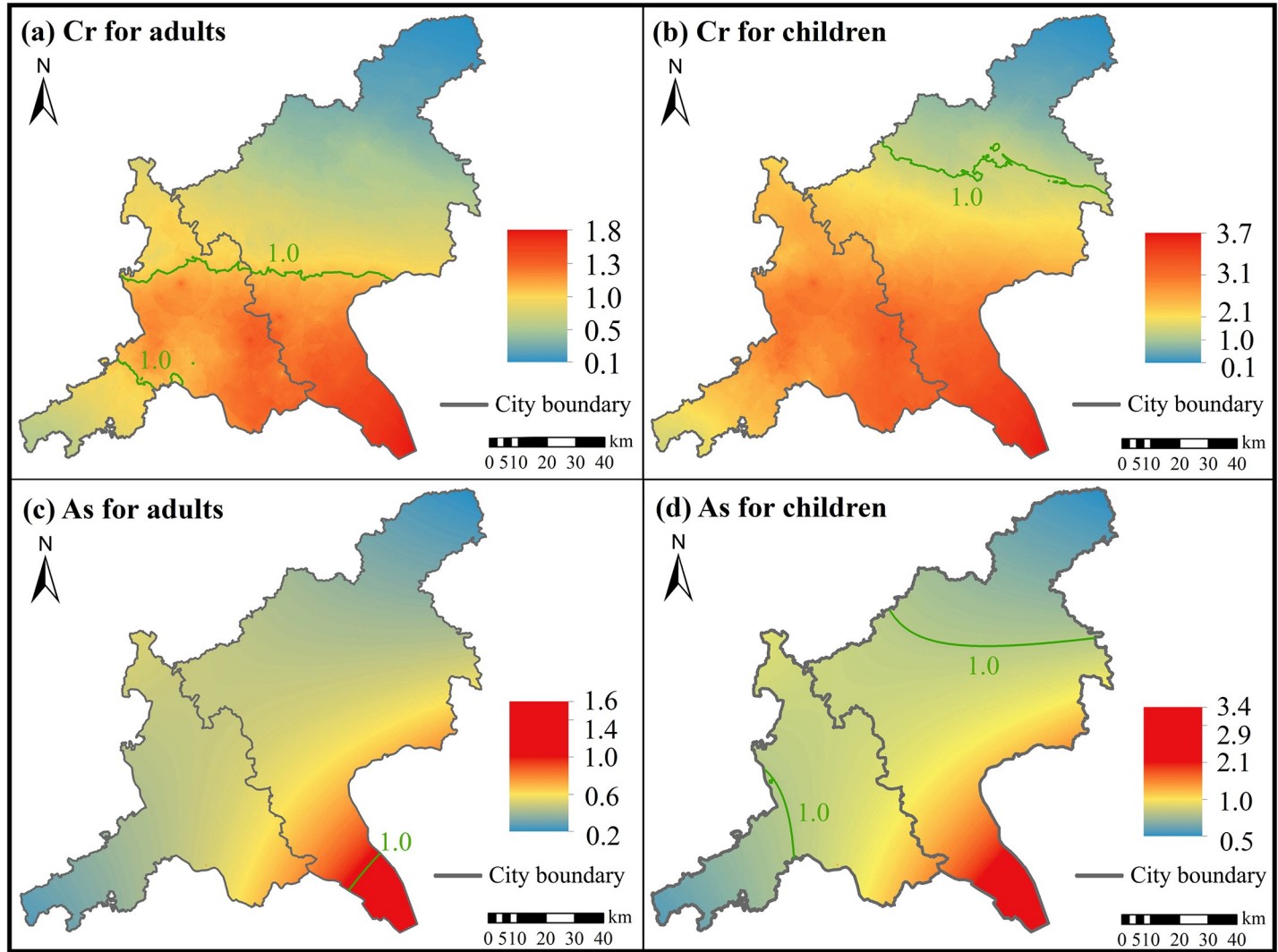

**Fig 6. Spatial distribution of estimated HQ values.** Cr for (a) adults and (b) children, As for (c) adults and (d) children.

The spatial distributions of HQ and CR of all interpolated heavy metals were similar for both adults and children. The highest HQ and CR values were distributed in the southeast regions of the study area (Figs 6 and 7), indicating that people living in the southeast were more likely to be exposed to potential health risks [53]. The HQ values of Cr and As greater than 1.0 were distributed in a small region and most regions of the study area for adults and children, respectively (Fig 6). The CR values of As greater than the maximum tolerable risk ($1\times10^{-4}$) only concentrated in a small region in the southeast region in the study area (Fig 7E). However, the CR values of Cd and Cr for both adults and children and the CR values of As for children were greater than the maximum tolerable risk ($1\times10^{-4}$) appeared in most regions of the study area (Fig 7A–7D and 7F).

## Conclusions

1. The average content of the measured 8 heavy metals in agricultural soil in the study area did not exceed the standard value. However, the average content of Cd, Cu, Hg and As

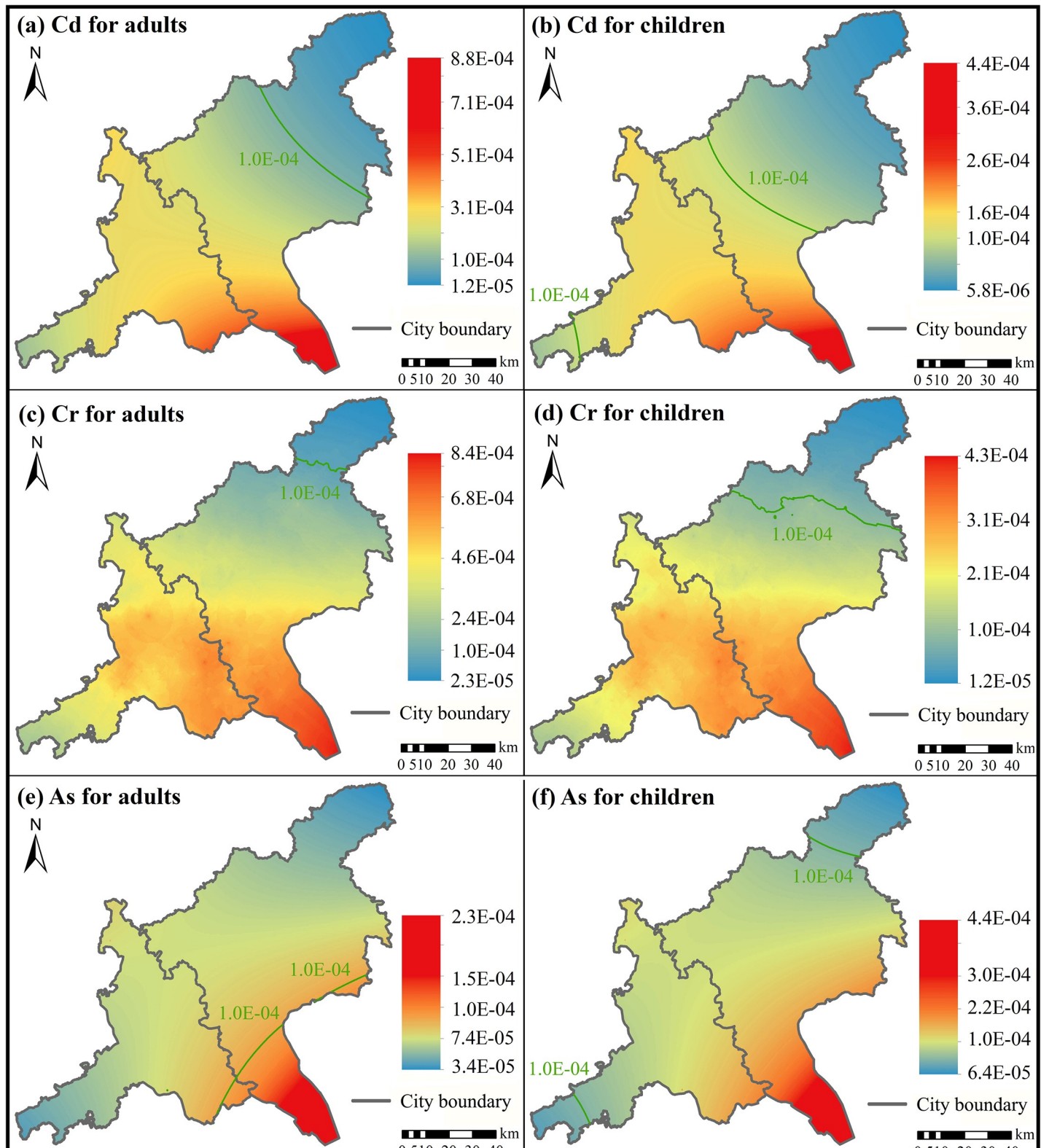

**Fig 7. Spatial distribution of CR values.** Cd for (a) adults and (b) children, Cr for (c) adults and (d) children, and As for (e) adults and (f) children.

were greater than their background values of Guangdong province. Relatively high heavy metal content was appeared at the surrounding areas of the city boundary between Guangzhou and Foshan due to the influence of man-made activities.

2. According to the results of the geoaccumulation index and the potential ecological risk, heavy metal pollution was mainly from Cd and Hg in agricultural soil of the study area. Most regions of Foshan city and a small region close to the Guangzhou-Foshan administrative boundary were suffered relatively high ecological risk.

3. Both adults and children in the study area were likely to experience non-carcinogenic and carcinogenic risks exposed through product consumption. The non-carcinogenic risks were mainly caused by Cr and As. The carcinogenic risks were mainly from Cr, Cd and As. Children in most regions of the study area might experience potential non-carcinogenic risks caused by Cr and As. Except As for adults, carcinogenic risks caused by Cd and Cr for both adults and children and As for children were greater than the maximum tolerable risk in most regions of the study area.

## Supporting information

**S1 File.**
(DOCX)

## Acknowledgments

The authors are also grateful to the reviewers and the Associate Editor for their constructive comments.

## Author Contributions

**Conceptualization:** Yi Xiao, Tingping Ouyang.

**Formal analysis:** Mingyan Guo.

**Methodology:** Yi Xiao.

**Software:** Yi Xiao.

**Writing – original draft:** Yi Xiao, Mingyan Guo, Xiaohong Li, Xixiang Luo, Ruikang Pan.

**Writing – review & editing:** Yi Xiao, Mingyan Guo, Tingping Ouyang.

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
