## [Decision Letter · Decision Letter 0]

14 Aug 2020

PONE-D-20-19278

Spatial distribution, pollution, and health risk assessment of heavy metal in agricultural surface soil for the Guangzhou-Foshan urban zone, South China

PLOS ONE

Dear Dr. Ouyang,

Thank you for submitting your manuscript to PLOS ONE. After careful consideration, we feel that it has merit but does not fully meet PLOS ONE’s publication criteria as it currently stands. Therefore, we invite you to submit a revised version of the manuscript that addresses the points raised during the review process.

We look forward to receiving your revised manuscript.

Kind regards,

Bing Xue, Ph.D.

Academic Editor

PLOS ONE

Journal Requirements:

3. Please clarify in your Methods section geographical coordinates of the dataset. Please also provide information on any field permits obtained for sampling, and if no field permits were obtained, a brief statement why.

4. Please include your tables as part of your main manuscript and remove the individual files. Please note that supplementary tables (should remain/ be uploaded) as separate "supporting information" files.

6. We note that Figures 1, 2, 4, 6, 7 in your submission contain map images which may be copyrighted. All PLOS content is published under the Creative Commons Attribution License (CC BY 4.0), which means that the manuscript, images, and Supporting Information files will be freely available online, and any third party is permitted to access, download, copy, distribute, and use these materials in any way, even commercially, with proper attribution. For these reasons, we cannot publish previously copyrighted maps or satellite images created using proprietary data, such as Google software (Google Maps, Street View, and Earth). For more information, see our copyright guidelines: http://journals.plos.org/plosone/s/licenses-and-copyright.

6.1.    You may seek permission from the original copyright holder of Figures 1, 2, 4, 6, 7 to publish the content specifically under the CC BY 4.0 license. 

6.2.    If you are unable to obtain permission from the original copyright holder to publish these figures under the CC BY 4.0 license or if the copyright holder’s requirements are incompatible with the CC BY 4.0 license, please either i) remove the figure or ii) supply a replacement figure that complies with the CC BY 4.0 license. Please check copyright information on all replacement figures and update the figure caption with source information. If applicable, please specify in the figure caption text when a figure is similar but not identical to the original image and is therefore for illustrative purposes only.

Reviewers' comments:

Reviewer's Responses to Questions

**Comments to the Author**

1. Is the manuscript technically sound, and do the data support the conclusions?

Reviewer #1: Yes

2. Has the statistical analysis been performed appropriately and rigorously? 

Reviewer #1: Yes

3. Have the authors made all data underlying the findings in their manuscript fully available?

Reviewer #1: Yes

4. Is the manuscript presented in an intelligible fashion and written in standard English?

Reviewer #1: Yes

5. Review Comments to the Author

Reviewer #1: The manuscript presents … on surface soil samples collected from agricultural regions with the Guangzhou-Foshan urban zone, aiming to evaluate heavy metal pollution and its potential impact on ecosystem and human health. My general view is that it is valuable and meaningful for agricultural product and public health for the study area. However, some minor revisions are needed before it can be accepted for publication in the Plos ONE.

1. Some expression were not suitable and some grammatical problems were appeared in the text. So the language should be improved.

2. The progresses concerned heavy metal pollution in agricultural soils and its human health risk assessment for the study area should be added in the Introduction part.

3. Since the formulas for human health risk assessment are not proposed or revised by the present study, they can be moved to supplementary materials.

4. Line 208 to 210, the ranges of heavy metal contents can be easily seen from Table 1. Therefore, they are not needed to be repeated in the text.

5. The authors only described the spatial distribution of heavy metal contents. In my opinion, the reasons caused these spatial distribution are also needed to be discussed.

6. Some discussion should be added to explain why stronger ecological risks were appeared at the region close to the city boundary and Foshan city.

7. Figure 1 is not clear enough, the resolution of the maps should be improved. And the “DEM” in the legend should be replaced by “altitude”.

8. In Figure 2, the units are consistent for all heavy metals, so the unit can be presented in captions. And the order numbers such as (a), (b), (c), etc. should be settled before the subtitles as (a) Cd, (b) Pb, (c) Cr, etc.

9. The description “No pollution - medium pollution”, “Medium pollution - heavy pollution”, and “Heavy pollution – very heavy pollution” in legend of Fig. 3a should be simplified to “No - medium pollution”, “Medium - heavy pollution”, and “Heavy – very heavy pollution”.

10. The “Boundary” within legends of Fig. 6 and Fig. 7 should be replaced by “City boundary”. The green lines stand for “1.0” and “1.0E-04” can be placed in the color scale.

6. PLOS authors have the option to publish the peer review history of their article (what does this mean?). If published, this will include your full peer review and any attached files.

Reviewer #1: **Yes: **Li Fangbai

---

## [Author Response · Author response to Decision Letter 0]

31 Aug 2020

Response to academic editor:

A:Thanks. We have adjusted the format of the manuscript in accordance with the format template of your journal.

2.We suggest you thoroughly copyedit your manuscript for language usage, spelling, and grammar. If you do not know anyone who can help you do this, you may wish to consider employing a professional scientific editing service. 

A: Thanks. Sheng Gao, a master majors in English Translation from the Zhongnan University of Economics and Law helped us to go through the manuscript for language usage.

3.Please clarify in your Methods section geographical coordinates of the dataset. Please also provide information on any field permits obtained for sampling, and if no field permits were obtained, a brief statement why.

A: Thanks. The geographical coordinates of the dataset is GCS WGS 1984. We also have added it to the M&M section of the revised manuscript (Lines 178 to 179) . Besides, we communicated with the landowners and got their permissions before sampling. So there is no other provide information.

4.Please include your tables as part of your main manuscript and remove the individual files. Please note that supplementary tables (should remain/ be uploaded) as separate "supporting information" files.

A: Thanks. The Tables were inserted after the first cited paragraph in the revised manuscript.

5.We note that you have stated that you will provide repository information for your data at acceptance. Should your manuscript be accepted for publication, we will hold it until you provide the relevant accession numbers or DOIs necessary to access your data. If you wish to make changes to your Data Availability statement, please describe these changes in your cover letter and we will update your Data Availability statement to reflect the information you provide.

A: Thanks. We will not change the data availability statement.

6.We note that Figures 1, 2, 4, 6, 7 in your submission contain map images which may be copyrighted. All PLOS content is published under the Creative Commons Attribution License (CC BY 4.0), which means that the manuscript, images, and Supporting Information files will be freely available online, and any third party is permitted to access, download, copy, distribute, and use these materials in any way, even commercially, with proper attribution. For these reasons, we cannot publish previously copyrighted maps or satellite images created using proprietary data, such as Google software (Google Maps, Street View, and Earth). For more information, see our copyright guidelines: http://journals.plos.org/plosone/s/licenses-and-copyright.

A: Thanks. The urban administrative boundaries used in Figures 1, 2, 4, 6, 7 in the manuscript are from the Department of Natural Resources of Guangdong Province (http://nr.gd.gov.cn/gdlr_public/map/3/gz/index.html). We used an ArcGIS to extract data through georeferencing and vectorization. Therefore, the data is scientific and reasonable and does not involve any copyright issues.

7.Please include captions for your Supporting Information files at the end of your manuscript, and update any in-text citations to match accordingly. Please see our Supporting Information guidelines for more information: http://journals.plos.org/plosone/s/supporting-information.

A: Thanks. We had followed this guide. 

Response to reviewer #1 (Comments to Author):

1. Some expression were not suitable and some grammatical problems were appeared in the text. So the language should be improved.

A: Thanks. Sheng Gao, a master majors in English Translation from the Zhongnan University of Economics and Law helped us to go through the manuscript for language usage. The unsuitable expression and grammatical problems were improved in the revised manuscript.

2. The progresses concerned heavy metal pollution in agricultural soils and its human health risk assessment for the study area should be added in the Introduction part.

A: Thanks. Some previous studies were cited to introduce the progresses concerned heavy metal pollution in agricultural soils and its human health risk assessment for the study area in the Introduction part of the revised manuscript (Lines 49-56, 76-77, 85-93). 

3.Since the formulas for human health risk assessment are not proposed or revised by the present study, they can be moved to supplementary materials.

A: Thanks. Based on the background conditions and particularities of the study area, we referred to Jiang, Y.X.'s formulas of the potential human health risk assessment model, and didn’t do any revision. And the formulas for human health risk assessment were moved to supplementary materials. 

4.Line 208 to 210, the ranges of heavy metal contents can be easily seen from Table 1. Therefore, they are not needed to be repeated in the text.

A: Thanks. These repeatable texts were deleted and “It can be clearly seen from the results listed in Table 1 that …” was used in the revised manuscript (Lines 246-247).

5.The authors only described the spatial distribution of heavy metal contents. In my opinion, the reasons caused these spatial distribution are also needed to be discussed.

A: Thanks. Comparing previous research results and analyzing the source of soil heavy metal content in the study area, we infer that there are two main reasons for the differences in the spatial distribution of soil heavy metals: one is the influence of natural factors, and the other is the influence of social activities. The study area is located in the alluvial plains of the Pearl River Delta. The terrain is high in the north and the south is densely covered in river networks. Therefore, the accumulation of heavy metals in the soil in the central and southern regions is relatively serious. At the same time, human activities are concentrated in areas with gentle terrain and abundant water resources. Emissions from a series of industrial and agricultural activities have led to high levels of heavy metal pollution in the soil in the southern part of the study area. This statements were added in Results and discussion section of the revised manuscript (Lines 212 to 217, 336 to 337).

6. Some discussion should be added to explain why stronger ecological risks were appeared at the region close to the city boundary and Foshan city.

A: Thanks. As mentioned above, topographical factors, hydrological factors and human social activities will cause the accumulation of heavy metals in the soil. When heavy metals in the soil accumulate to a certain level, it will cause pollution. Therefore, the higher the content of heavy metals in the soil, the greater the risk of pollution. Based on the calculation process of potential ecological risk, the high level of potential ecological risk is affected by the soil heavy metal content and toxicity coefficient. From the above description of the spatial distribution of soil heavy metal content, it can be seen that the high-risk pollution at the junction of Guangzhou - Foshan urban area is mainly due to the high soil heavy metal content (Lines 250-251, 342 of the revised manuscript).

7.Figure 1 is not clear enough, the resolution of the maps should be improved. And the “DEM” in the legend should be replaced by “altitude”.

A: Thanks, the resolution of the maps were improved and the “DEM” was replaced by “Elevation”.

8.In Figure 2, the units are consistent for all heavy metals, so the unit can be presented in captions. And the order numbers such as (a), (b), (c), etc. should be settled before the subtitles as (a) Cd, (b) Pb, (c) Cr, etc.

A: Thanks. Revised.

9.The description “No pollution - medium pollution”, “Medium pollution - heavy pollution”, and “Heavy pollution – very heavy pollution” in legend of Fig. 3a should be simplified to “No - medium pollution”, “Medium - heavy pollution”, and “Heavy – very heavy pollution”.

Thanks. Revised.

10.The “Boundary” within legends of Fig. 6 and Fig. 7 should be replaced by “City boundary”. The green lines stand for “1.0” and “1.0E-04” can be placed in the color scale.

A: Thanks. “Boundary” was replaced by “City boundary”. The values 1.0 and 1.0E-04 were marked next to the green lines in Fig 6 and Fig 7.

---

## [Decision Letter · Decision Letter 1]

9 Sep 2020

Spatial distribution, pollution, and health risk assessment of heavy metal in agricultural surface soil for the Guangzhou-Foshan urban zone, South China

PONE-D-20-19278R1

Dear Dr. Ouyang,

We’re pleased to inform you that your manuscript has been judged scientifically suitable for publication and will be formally accepted for publication once it meets all outstanding technical requirements.

Kind regards,

Bing Xue, Ph.D.

Academic Editor

PLOS ONE

Additional Editor Comments (optional):

Reviewers' comments:

Reviewer's Responses to Questions

**Comments to the Author**

1. If the authors have adequately addressed your comments raised in a previous round of review and you feel that this manuscript is now acceptable for publication, you may indicate that here to bypass the “Comments to the Author” section, enter your conflict of interest statement in the “Confidential to Editor” section, and submit your "Accept" recommendation.

Reviewer #1: All comments have been addressed

2. Is the manuscript technically sound, and do the data support the conclusions?

Reviewer #1: Yes

3. Has the statistical analysis been performed appropriately and rigorously? 

Reviewer #1: Yes

4. Have the authors made all data underlying the findings in their manuscript fully available?

Reviewer #1: Yes

5. Is the manuscript presented in an intelligible fashion and written in standard English?

Reviewer #1: Yes

6. Review Comments to the Author

Reviewer #1: All questions were carefully addressed one by one. The quality of the manuscript had been improved greatly. Acceptable.

7. PLOS authors have the option to publish the peer review history of their article (what does this mean?). If published, this will include your full peer review and any attached files.

Reviewer #1: No

---

## [Editor Report · Acceptance letter]

29 Sep 2020

PONE-D-20-19278R1 

Spatial distribution, pollution, and health risk assessment of heavy metal in agricultural surface soil for the Guangzhou-Foshan urban zone, South China 

Dear Dr. Ouyang:

I'm pleased to inform you that your manuscript has been deemed suitable for publication in PLOS ONE. Congratulations! Your manuscript is now with our production department. 

Kind regards, 

on behalf of

Professor Bing Xue 

Academic Editor

PLOS ONE